# Indirect and Direct 65+ Patient Reporting of Non-Steroidal Anti-Inflammatory Drug-Induced Adverse Drug Reactions as a Source of Information on Polypharmacy and Polypharmacy-Related Risk

**DOI:** 10.3390/medicina59091585

**Published:** 2023-08-31

**Authors:** Kamila Sienkiewicz, Monika Burzyńska, Izabela Rydlewska-Liszkowska, Jacek Sienkiewicz, Ewelina Gaszyńska

**Affiliations:** 1Department of Management and Logistics in Healthcare, Medical University of Lodz, Lindleya Street 6, 90-131 Lodz, Poland; izabela.rydlewska-liszkowska@umed.lodz.pl (I.R.-L.); jacek.sienkiewicz@stud.umed.lodz.pl (J.S.); 2Department of Epidemiology and Biostatistics, Medical University of Lodz, Żeligowskiego Street 7, 990-752 Lodz, Poland; monika.burzynska@umed.lodz.pl; 3Department of Nutrition and Epidemiology, Medical University of Lodz, Żeligowskiego Street 7, 990-752 Lodz, Poland; ewelina.gaszynska@umed.lodz.pl

**Keywords:** adverse reaction, polypharmacy, patient reporting, EudraVigilance, pharmacovigilance, safety monitoring, drug interactions

## Abstract

*Background and Objectives*: Non-steroidal anti-inflammatory drugs (NSAIDs), which have anti-inflammatory and analgesic properties, are commonly used in the treatment of various, particularly frequent, as well as chronic, conditions in older patients. Due to common polypragmasia in these patients and a high risk of adverse drug reactions (ADRs) and drug interactions, pain management poses a therapeutic challenge. This study describes the importance of ADR reports in the identification of polypharmacy and the ensuing interactions. *Materials and Methods*: Both healthcare professionals (HPs) and non-healthcare professionals (non-HPs) reports collected in the EudraVigilance database of NSAIDs, including most commonly co-reported medications and reported reactions, were analysed and differences between HPs and non-HPs reports were identified. *Results*: In the analysed period and group, non-HPs reported more reactions but indicated fewer drugs as suspect or concomitant. The outcomes of our analysis indicate more HP engagement and more detailed reports of serious ADRs when compared to non-serious individual case safety reports (ICSRs) by non-HPs, which appeared more detailed. Such reactions as kidney failure and increased risk of bleeding are known adverse reactions to NSAIDs and common symptoms of their interactions, which were described in the available literature. They were much more frequently reported by HPs than by non-HPs. Non-HPs more frequently reported reactions that may have been considered less significant by HPs. *Conclusions*: The differences between healthcare professionals’ (HPs) and non-healthcare professionals’ (non-HPs) reports may result from the fact that the reports from patients and their caregivers require a professional medical diagnosis based on symptoms described by the patient or additional diagnostic tests. This means that when appropriately classified, medically verified, and statistically analysed, the data may provide new evidence for the risks of medication use or drug interactions.

## 1. Introduction

Polypharmacy, which is defined as the simultaneous use of more than five medicinal products, is a significant problem that concerns the elderly in particular. It may lead to many drug interactions and adverse drug reactions (ADRs), which result in, among others, health problems, ineffective treatment, decreased quality of life, or additional costs for the health care system [1,2]. The risk of ADRs increases with a higher number of drugs used, which is more frequent and perilous in the elderly, who often suffer from multiple comorbidities [3].

Although there are guidelines for polypharmacy management in the elderly, they are very complex, which affects their practical application. To date, polypharmacy management has resulted in decision support systems to automate the implementation of guidelines and provide additional information on medications [2]. Taking into account the increased incidence of rheumatic diseases in the elderly, it is crucial for healthcare providers to be aware of the potential treatment-related benefits and risks. Polypharmacy and polymorbidity are further potential challenges for pain management and rheumatoid disease treatment in elderly patients [4,5].

Pain, a frequent symptom observed in clinical practice, particularly affects elderly and chronically ill patients. As polypharmacy is very common in such patients, pharmacological treatment of pain becomes a significant therapeutic challenge. A particularly high risk of ADRs and drug interactions applies to non-steroid anti-inflammatory drugs (NSAIDs) [6]. They are used on a permanent basis by approx. 5% of the population, whereas this percentage increases to 10–20% in patients aged over 65 years. The problem of NSAID safety is particularly important in the elderly because of the wide use of NSAIDs, the numerous cases of ADRs, which are influenced by both patients’ age and health, and finally, the incidence of polypharmacy and thus of drug interactions [7].

This study describes the importance of ADR reports in the identification of polypharmacy and the consequent drug interactions, as well as analysing the share of reports from patients and their caregivers in the identification of potential risks of NSAID interactions with other medications. The primary source of an individual case safety report (ICSR) is an adverse reaction-reporting person, including a healthcare professional (HP), defined as a person with medical qualifications (e.g., a physician, a dentist, a pharmacist, or a nurse), or a non-healthcare professional (non-HP), defined as a consumer (a patient, a patient’s relative, or a caregiver). In our analysis, each ICSR included at least one medicinal product characterised by the reporter (HP or non-HP) as a drug suspected of causing an adverse reaction. ICSR could additionally include information about concomitant drug/-s reported that were used by patients but were not reported as suspect ones. 

The aim of this study was to analyse the impact of ADR reports on the identification of polypharmacy-related risk in the elderly and the consequent drug interactions by comparing and contrasting the reports from both HPs and non-HPs. 

This study compared and contrasted the reports from both reporting groups as a source of information about polypharmacy and potential drug interactions by analysing the most frequently co-reported drugs and reported interactions.

Another objective of this study was also to compare reports of serious vs. non-serious adverse reactions with regard to the mean number of reactions and the mean number of drugs reported in a single case based on the reports from HPs and non-HPs. 

Moreover, this study also assessed the correlation between the reported number of medications and reactions in serious and non-serious ADRs reported by non-HPs and HPs.

This study analysed individual case safety reports collected in the EudraVigilance database of non-steroidal anti-inflammatory drugs. In July 2012, EU legislation empowered non-HPs, i.e., patients and their caregivers, to report any suspected adverse reactions, which made spontaneous reporting by non-HPs an additional source of data in the EU pharmacovigilance database [8,9]. 

The reports included in our analysis concerned patients aged over 65, as this age group uses multiple medicinal products and presents multiple comorbidities in their medical histories.

In order to analyse interactions, which are of particular importance in elderly patients because of the growing problem of polypharmacy, only the reports that included a total of at least five reported medicinal substances were eventually selected for investigation. 

## 2. Materials and Methods

### 2.1. Materials

The analysed data were collected from the following website: www.adrreports.eu (accessed on 13 April 2022), which provides public access to aggregated EudraVigilance data [10]. The EudraVigilance database is an electronic system designed to collect and analyse data on medicinal product safety, and all reported cases of suspected adverse reactions to medications authorised in the European Economic Area (EEA) are transferred to the database.

The following inclusion criteria were used in the analysis:ICSRs collected in EudraVigilance.ICSRs are related to substances that belong to the M01A anti-inflammatory and anti-rheumatic products, non-steroids group, according to the Anatomical Therapeutic Chemical (ATC) Classification.Reactions are classified as serious or non-serious.Reactions assigned to all geographic origin groups.Reactions reported by HPs and non-HPs.ICSRs related to all patient sex groups.Reports for patients aged >65.ICSRs related to all reaction groups.Reports are assigned to all reported suspected reactions.ICSRs sent to the EudraVigilance database between 2018 and 2021.

Our analysis included individual case safety from EudraVigilance from 2018 until the end of 2021. The selected timespan was determined because both serious and non-serious adverse reactions to medicinal products have had to be obligatorily reported to EudraVigilance since November 2017. The new legislation simplified suspected ADR reporting in Europe and increased the transparency and availability of collected data, which improved the data analysis as a result [11,12,13]. Our analysis included reactions reported by both females and males as well as those assigned to all geographic origin groups, which means that the investigated reports came from both European Economic Area (EEA) and non-EEA countries. Consequently, the results of this study are applicable to a wide population. 

The European Medicines Agency presents an overview of data management and quality assurance activities performed on information about suspected adverse reactions that is presented in the EudraVigilance database [14]. The Agency collaborates with other stakeholders (marketing authorisation and Member States) to ensure the integrity and quality of the information collected in the database. The Agency summarizes quality assurance activities to ensure consistent, complete, correct, and well-structured information submitted in ICSRs. 

### 2.2. Methods 

The data extraction and processing were performed using custom Python scripts and Pandas library [15]. SciPy Python library was used to perform statistical tests [16]. The results were visualised using Matplotlib and Seaborn Python libraries [17,18].

#### 2.2.1. Statistical Analysis

The observed data on the number of ‘reactions’, ‘suspect’, and ‘concomitant’ drugs are discrete, count-type, and non-normally distributed. To determine statistically significant differences between the groups, a non-parametric Mann–Whitney U test was used, which does not make any assumptions about the distribution of data. The obtained Mann–Whitney test U statistic was then used to calculate the rank-biserial correlation [19] in order to estimate the effect size [20]. Furthermore, the bootstrapping method was used to estimate the mean difference between samples and the 95% confidence interval of the mean difference between samples. Ten thousand bootstraps meant that different replicates were taken. Resamples of size equal to each group size were drawn with replacements from the datasets. The 2.5th and the 97.5th percentiles of the ranked differences form the boundaries of the 95% confidence interval.

The characteristics of the medicinal products as suspect or concomitant are provided by the primary source (HP or non-HP) and reported by the sender (regulatory authority or Marketing Authorisation Holder. 

Medicinal product means any substance or combination of substances that can be used to treat or prevent disease, that may restore, correct, or modify physiological functions via a pharmacological, immunological, or metabolic impact, or that may be used to make a medical diagnosis [8]. Suspect drug is a medicinal product suspected of causing an adverse drug reaction, according to the reporter. Based on the minimum criteria for reporting suspected adverse reactions, one or more suspect medicinal products need to be present. If reporters provided more data about a medicinal product that they did not suspect of having an adverse drug reaction, the drug is described as concomitant. 

#### 2.2.2. Drug Reaction Analysis

The names of reactions were extracted from the Reaction List. An analysis of drug reactions was performed. First, the number of reactions within the merged reaction lists was counted to estimate the incidence of reactions. The obtained figures were then divided by the number of case reports, separately for HPs and non-HPs, and then additionally, separately for serious and non-serious ADR reports. The obtained ratio can indicate how often a given reaction was reported in patients in the analysed groups. The two-sample Z-test was used to check the statistical significance of the difference. 

Adverse drug reaction, or adverse reaction, is defined as a response to a medicinal product that is noxious and unintended [8].

#### 2.2.3. Statistical Analysis of Potential Interaction Incidence

In order to investigate drug interactions, the reports that contained at least 5 reported medicinal substances were included in the analysis.

The drug names from the ‘suspect’ and ‘concomitant’ drug lists were extracted, merged, and collapsed to obtain a list of unique drug names. Then the list was automatically curated to replace different names of the active compound with one name (e.g., vitamin D, cholecalciferol, and ergocalciferol). The primary drug was removed from the list.

The dataset was then filtered to obtain a new dataset of cases with five or more ‘suspect’ and ‘concomitant’ drugs. The drug interaction analysis was performed for five primary drugs with the highest number of cases (ibuprofen, diclofenac, celecoxib, naproxen, and meloxicam). First, how frequently the drug appeared within the merged drug lists was counted to estimate the frequency of interacting drugs. The obtained figures were then divided by the number of cases for each primary drug, separately for HP and non-HP groups. The obtained ratio can indicate how often a given additional drug was used in patients receiving the primary drug. The two-sample Z-test was used to check whether the differences were statistically significant. 

## 3. Results

### 3.1. Drug Interaction Analysis

In the analysed reports, there were 4240 cases where the total of suspect and concomitant drugs was equal to or higher than five; these included 3447 reported by HPs and 793 reported by non-HPs. 

Figure 1 shows the number of cases of specific primary drugs used. 

Given the number of cases, the interaction analysis was performed only for the following drugs: ibuprofen, diclofenac, celecoxib, naproxen, and meloxicam.

In order to estimate the frequency of interacting drugs, first, how often a drug appeared within merged ‘suspect’ and ‘concomitant’ drug lists was counted. The obtained figures were then divided by the number of cases for each primary drug. This was performed separately for the HP and non-HP groups. The obtained ratio can be interpreted as how often a given additional drug was reported in patients receiving the primary drug.

Out of 20 drugs reported jointly with other medications, ibuprofen was more frequently reported by non-HPs compared to HPs, by 16.5 percentage points (pp) *p* = 1.21 × 10^−11^ when reported in combination with chlorphenamine, by 14.85 pp, *p* = 8.99 × 10^−10^ when with pseudoephedrine, by 7.90 pp, *p* = 1.00 × 10^−9^ with leflunomide, by 7.64 pp, *p* = 2.16 × 10^−8^ with adalimumab, by 7.54 pp, *p* = 3.25 × 10^−6^ with duloxetine, by 6.92 pp, *p* = 1.00 × 10^−9^ with naproxen, by 6.67 pp, *p* = 7.53 × 10^−4^ with candesartan, by 6.42 pp, *p* = 1.53 × 10^−3^ with methotrexate, and by 6.19 pp, *p* = 5.23 × 10^−3^ with ascorbic acid (Figure 2).

Among the 20 drugs most frequently reported by HPs in combination with other drugs, ibuprofen was more frequently reported by HPs compared to non-HPs by 14.72 pp, *p* = 6.61 × 10^−5^ in combination with rivaroxaban, by 12.98 pp, *p* = 1.50 × 10^−4^ with pantoprazole, by 10.77 pp, *p* = 2.94 × 10^−4^ when with potassium chloride, and by 9.98 pp, *p* = 3.36 × 10^−3^ with furosemide (Figure 3).

In the group of 20 drugs that were mosyes it is t frequently reported by non-HPs in combination with diclofenac (Figure 4), non-HPs more frequently than HPs reported the use of hydrochlorothiazide (by 8.56 pp; *p* = 0.0124), vitamin D (by 8.28 pp; *p* = 8.01 × 10^−3^), ascorbic acid (by 7.04 pp; *p* = 1.83 × 10^−3^), ibuprofen (by 6.54 pp; *p* = 0.02), losartan (by 4.52 pp; *p* = 0.017), pregabalin (by 4.34 pp; *p* = 0.0426), and irbesartan (by 4.12 pp; *p* = 0.216), but more rarely with pantoprazole (by 8.87 pp; *p* = 0.016).

The analysis of 20 drugs most frequently reported by HPs in combination with diclofenac (Figure 5) showed that HPs more frequently than non-HPs reported the use of pantoprazole (by 8.87 pp; *p* = 0.016), folic acid (by 8.78 pp; *p* = 3.15 × 10^−3^), furosemide (by 8.78 pp; *p* = 4.22 × 10^−3^), lisinopril (by 7.61 pp; *p* = 3.768 × 10^−3^), and hydroxychloroquine (by 7.06 pp; *p* = 0.018), whereas more infrequently with hydrochlorothiazide (by 8.56 pp; *p* = 0.0124) and vitamin D (by 8.28 pp; *p* = 8.01 × 10^−3^).

Among the 20 drugs most frequently reported by non-HPs in combination with celecoxib (Figure 6), non-HPs more frequently than HPs reported vitamin D (by 11.75 pp; *p* = 2.19 × 10^−4^), levothyroxine (by 8.94 pp; *p* = 0.065), omeprazole (by 6.33 pp; *p* = 1.58 × 10^−2^), as well as pantoprazole (by 6.02 pp; *p* = 0.011), gabapentin (by 5.87 pp; *p* = 8.91 × 10^−3^), oxycodone (by 5.43 pp; *p* = 0.014), and hydroxychloroquine (by 5.64 pp; *p* = 6.12 × 10^−2^).

When we analysed the 20 drugs most frequently reported by HPs in combination with celecoxib (Figure 7), it appeared that HPs more frequently than non-HPs reported the use of rebamipide (by 9.79 pp; *p* = 5.97 × 10^−5^), leflunomide (by 9.04 pp; *p* = 8.05 × 10^−4^), and magnesium (by 7.31 pp; *p* = 4.09 × 10^−3^), whereas vitamin D, levothyroxine, omeprazole, and hydroxychloroquine were reported more infrequently than by non-HPs (by 11.7 pp; *p* = 2.19 × 10^−4^, 8.95 pp; *p* = 7.35 × 10^−4^, 6.33 pp; *p* = 1.58 × 10^−2^, and 5.64 pp; *p* = 6.12 × 10^−2^, respectively).

The analysis of the 20 drugs that were most often reported by HPs in combination with naproxen (Figure 8) revealed that HPs more frequently than non-HPs reported the use of rivaroxaban (by 9.2 pp; *p* = 1.40 × 10^−4^), codeine (by 9.07 pp; *p* = 1.23 × 10^−3^) and bisoprolol (by 7.48 pp; *p* = 1.85 × 10^−3^), whereas non-HPs more frequently than HPs co-reported naproxen with esomeprazole (by 47.66 pp; *p* = 1.32 × 10^−33^), with pantoprazole (by 36.03 pp; *p* = 9.22 × 10^−22^), omeprazole (by 32.68 pp; *p* = 8.28 × 10^−15^ × 10^−12^), with lansoprazole (by 28.43% *p* = 5.26 × 10^−13^), acetylsalicylic acid (by 15.00 pp; *p* = 8.97 × 10^−5^), levothyroxine (by 8.55 pp; *p* = 7.25 × 10^−3^), amlodipine (by 8.01 pp; *p* = 0.015), simvastatin (by 6.34 pp; *p* = 0.038), and tramadol (by 5.21 pp; *p* = 0.049).

Among the 20 drugs most frequently reported by non-HPs in combination with naproxen (Figure 9), in addition to the drugs listed above, i.e., esomeprazole, pantoprazole, omeprazole, lansoprazole, acetylsalicylic acid, levothyroxine, amlodipine, and simvastatin, the list also included rabeprazole (by 43.65 pp; *p* = 1.38 × 10^−35^), dexlansoprazole (by 38.33 pp; *p* = 3.60 × 10^−30^), lisinoprol (by 14.63 pp; *p* = 5.53 × 10^−7^), hydrochlorothiazide (by 8.93 pp; *p* = 0.0012), clopidogrel (by 11.50 pp; *p* = 2.55 × 10^−6^), azithromycin (by 15.32 pp; *p* = 2.35 × 10^−13^), and metoprolol (by 7.47 pp; *p* = 3.88 × 10^−3^).

When analysing the 20 drugs most frequently co-reported by HPs with meloxicam, a more frequent co-reporting of furosemide (by 12.31 pp; *p* = 0.048) compared to non-HPs was observed. No statistically significant differences were recorded among the 20 drugs most frequently co-reported by non-HPs in combination with meloxicam compared to the HPs group.

### 3.2. Drug Reaction Analysis

Reactions reported by HPs and non-HPs were varied. The number of different preferred terms (PTs) according to MedDRA is 2461 for HPs in comparison to 1617 PTs for non-HPs. It can therefore be assumed that although HPs reported fewer reactions on average, their diversity was greater compared to non-HPs. 

When analysing the reactions that were reported with different frequencies by HPs and non-HPs (Figure 10), the results of the two-sample Z-test for proportions showed the biggest absolute value of a difference between the reaction ratio for both analysed groups for gastrointestinal haemorrhage (by 4.48 pp; *p* = 2.98 × 10^−26^), acute kidney injury (AKI) (by 4.32 pp; *p* = 1.04 × 10^−24^), drug ineffectiveness (by 2.95 pp; *p* = 2.66 × 10^−11^), melaena (by 2.89 pp; *p* = 6.37 × 10^−20^), chronic kidney disease (by 2.67 pp; *p* = 1.22 × 10^−34^), upper gastrointestinal haemorrhage (by 2.57 pp; *p* = 1.03 × 10^−17^), product use in an unapproved indication (by 2.36 pp; *p* = 7.64 × 10^—18^), headache (2.34 pp; *p* = 8.56 × 10^−15^), drug interaction (2.22 pp; *p* = 1.80 × 10^−12^), (angioedema 2.08 pp; *p* = 4.25 × 10^−15^), and pain 2.08 pp; *p* = 1.07 × 10^−7^).

However, HPs more frequently than non-HPs (green colour on Figure 10) reported: gastrointestinal haemorrhage (by 4.48 pp; *p* = 2.98 × 10^−26^), AKI (by 4.32 pp; *p* = 1.04 × 10^−24^), melaena (2.89 pp; *p* = 6.37 × 10^−20^), upper gastrointestinal haemorrhage (by 2.57% *p* = 1.03 × 10^−17^), drug interaction (2.22 pp; *p* = 1.80 × 10^−12^), and angioedema (2.08 pp; *p* = 4.25 × 10^−15^). Non-HPs, in turn, reported more frequently than HPs (blue colour on Figure 10): drug ineffectiveness (by 2.95 pp; *p* = 2.66 × 10^−11^), chronic kidney disease (by 2.67 pp; *p* = 1.22 × 10^−34^), product use in unapproved indications (by 2.36 pp; *p* = 7.64 × 10^−18^), headache (2.34 pp; *p* = 8.56 × 10^−15^), and pain (by 2.08 pp; *p* = 1.07 × 10^−7^). 

The analysis of twenty reactions most frequently reported by non-HPs (Figure 11) and HPs (Figure 12) sorted according to statistically significant absolute pp values of the difference between reaction ratios revealed that for non-HPs, for all reactions, a preponderance of reporting was noted in favour of non-HP (blue colour), whereas for the group of twenty reactions most frequently reported by HPs, half of them (seven out of fourteen reactions) were more frequently reported by HPs (green colour) than by non-HPs. 

Serious ADRs (Figure 13) were reported with a different frequency by HPs compared to non-HPs. The results of a two-sample Z-test for proportions showed the biggest difference (absolute value) for the reaction ratios among the analysed groups for gastrointestinal haemorrhage (by 5.59 pp; *p* = 2.37 × 10^−22^), AKI (by 5.36 pp; *p* = 8.00 × 10^−21^), chronic kidney disease (by 3.96 pp; *p* = 2.20 × 10^−40^), melaena (by 3.6 pp; *p* = 4.35 × 10^−17^), drug ineffectiveness (by 3.28 pp; *p* = 2.51 × 10^−9^), and upper gastrointestinal haemorrhage (by 3.27 pp; *p* = 1.51 × 10^−15^).

HPs more frequently than non-HPs reported the following serious ADRs (green colour on Figure 13): gastrointestinal haemorrhage (by 5.59 pp; *p* = 2.37 × 10^−22^), AKI (by 5.36 pp; *p* = 8.00 × 10^−21^), melaena (by 3.6 pp; *p* = 4.35 × 10^−17^), upper gastrointestinal haemorrhage (by 3.27 pp; *p* = 1.51 × 10^−15^), drug interaction (by 2.53 pp; *p* = 2.12 × 10^−10^), angioedema (by 2.52 pp; *p* = 1.11 × 10^−12^), and anaemia (by 2.43 pp; *p* = 6.19616 × 10^−10^). 

The reactions more often reported by non-HPs (blue colour on Figure 13) included: chronic kidney disease (by 3.96 pp; *p* = 2.20 × 10^−40^), drug ineffectiveness (by 3.28 pp; *p* = 2.51 × 10^−9^), product use in an unapproved indication (by 2.67 pp; *p* = 9.96 × 10^−15^), renal failure (by 2.57 pp; *p* = 8.21 × 10^−14^), pruritus (by 2.42 pp; *p* = 2.47 × 10^−7^), headache (by 2.28 pp; *p* = 7.06 × 10^−11^), pain (by 2.19 pp; *p* = 7.96 × 10^−6^), and peripheral swelling (by 2.02 pp; *p* = 1.15 × 10^−9^). 

The analysis of serious reactions most frequently reported by non-HPs (Figure 14) and HPs (Figure 15) sorted according to statistically significant absolute pp values of the difference between reaction ratios revealed that for the= non-HPs group, all of the first twenty reactions were more often reported by non-HPs (blue colour), whereas for the HPs group, half of them (seven out of fourteen reactions) were more often reported by HPs (green colour). These reactions significantly overlapped with reactions for the analysed group in general (Figure 11 and Figure 12). 

Non-serious ADRs were reported (Figure 16) with varying frequency by HPs compared to non-HPs. The results of the two-sample Z-test for proportions showed the biggest difference (absolute value) for differences between the reaction ratios for both analysed groups for: rash (by 4.09 pp; *p* = 2.97 × 10^−5^), urticaria (by 4.07 pp; *p* = 2.73 × 10^−7^), drug ineffectiveness (by 2.74 pp; *p* = 8.34 × 10^−5^), off-label use (by 2.50 pp; *p* = 2.56 × 10^−9^), headache (by 2.42 pp; *p* = 6.0232 × 10^−5^), pain (by 2.33 pp; *p* = 1.35 × 10^−4^), fatigue (by 2.33 pp; *p* = 6.45 × 10^−7^), and arthralgia (by 2.07 pp; *p* = 9.76 × 10^−8^).

With regard to non-serious ADRs, HPs more frequently reported: rash (by 4.09 pp; *p* = 2.97 × 10^−5^) and urticaria (by 4.07 pp; *p* = 2.73 × 10^−7^), whereas non-HPs more often reported: drug ineffectiveness (by 2.74 pp; *p* = 8.34 × 10^−5^), off-label use (by 2.50 pp; *p* = 2.56 × 10^−9^), headache (by 2.42 pp; *p* = 6.0232 × 10^−5^), pain (by 2.33 pp; *p* = 1.35 × 10^−4^), fatigue (by 2.33 pp; *p* = 6.45 × 10^−7^), and arthralgia (by 2.07 pp; *p* = 9.76 × 10^−8^).

When analysing the statistically significant absolute pp values in the difference between reaction ratios, we observed that for the first 20 non-serious ADRs reported by non-HPs (Figure 17), all reactions except rash occurred more often in the non-HP group (blue colour). Contrastingly, the most popular reactions noticed in the all-cases group and the serious-cases group, such as gastrointestinal haemorrhage or kidney injuries, were not reported in the first 20 serious reactions reported by HPs (Figure 18).

### 3.3. Descriptive Statistics

Table 1 presents the data on the mean number of reactions, mean number of suspected drugs, and concomitant drugs reported between 2018 and 2021 by HPs and non-HPs.

The mean number of reactions in a single ICSR provided by non-HPs was higher than those from HPs, both in the analysed group in general and for serious and non-serious ADR reports analysed separately. In the investigated period, non-HPs reported more “reactions” (mean by approx. 0.395/case) and less “suspect” (mean by 0.388/case), “concomitant” (mean by 0.625/case), and “suspect_and_concomitant” (mean by 1013/case) drugs (Figure 19). The HPs/Non-HPs mean difference bootstrapping was + 95% CI, and the obtained results are statistically significant.

For the purpose of post-hoc analysis and correlation analysis, four sub-groups were created:Group 1—Healthcare Professionals and non-serious casesGroup 2—Healthcare Professionals and serious casesGroup 3—Non-Healthcare Professionals and non-serious casesGroup 4—Non-Healthcare Professionals and serious cases

A post-hoc test to assess statistically significant differences between groups indicated significance for all pairs, which means that the mean number of reactions differed significantly between the groups: 1 and 2 (2.03 and 3.50, respectively), 1 and 3 (2.03 and 2.56, respectively), 1 and 4 (2.03 and 3.99, respectively), and also 2 and 3, 2 and 4, and 3 and 4 (mean values in Table 1). The *p*-values are specified in Table 2. 

The post-hoc analysis indicated that the mean number of reactions in Group 3 was statistically significantly higher by 0.53 than in Group 1, whereas in Group 4, the value was higher by 0.5 than in Group 2, which means that non-HPs reported more reactions in both serious and non-serious cases.

The same analysis was conducted for the suspect_drug_cnt variable. The post-hoc test results were statistically significant, but not for all the pairs. That means that the mean level of suspect_drug_cnt differed significantly between the following groups: 1 and 2; 1 and 4; 2 and 3; 2 and 4; as well as 3 and 4 (mean values, Table 3). The *p*-values are specified in Table 3. 

A post-hoc analysis showed that the mean level of suspect_drug_cnt in Group 2 was higher by approx. 0.34 than in Group 4, thereby indicating that in serious ADR reports, HPs reported more suspect drugs (mean by approx. 0.34) than non-HPs. No statistically significant differences were observed in the mean value of suspect_drug_cnt between Groups 1 and 3, which means that non-serious ADRs reported comparatively the same number of drugs as HPs and non-HPs.

### 3.4. Correlation Analysis

#### 3.4.1. Correlation Analysis for the Reaction Count and Suspect Count Variables

When analysing all the data, the number of reported drugs correlated with the number of reported reactions. It was a statistically significant positive correlation with significant power (r = 0.4902, *p* = 0.00). Therefore, the number of reported reactions increased along with the increase in the number of reported drugs. 

In the analysis of the correlations between groups, the following results were obtained:

A statistically significant positive correlation between the number of reported substances and the number of reactions was observed in all groups. The number of reactions rose along with the increase in the number of drugs.

The most powerful correlation was noted for Group 2, with a correlation coefficient of 0.5395, which demonstrates its large power. The correlation of moderate power was observed for Group 4 (r = 0.3093 *p* = 0.00), and the weakest correlation was recorded in Groups 3 and 1 (coefficients of 0.1039 *p* = 0.001 and 0.944 *p* = 0.00, respectively).

The weaker correlation recorded between Group 2 and Group 4 may indicate decreased engagement of healthcare professionals in non-serious ADR reporting, whereas in serious ADR reports, a positive correlation was observed with a higher power for HPs than for non-HPs, which implies a higher detail orientation and engagement of HPs in reporting serious ADRs. 

#### 3.4.2. A Correlation Analysis of Reaction Count and Suspect and Concomitant Count Variables

A statistically significant positive correlation was observed between reaction count, suspect, and concomitant count in three out of four analysed groups (Groups 2, 3, and 4); an increase in the number of reported medications increased with a rise in the number of reactions.

The most powerful correlation was noted for Group 2, with a correlation coefficient of 0.4605 *p* = 0.00, which indicates its moderate power. A moderately powerful correlation was also recorded in Group 4 (r = 0.3219, *p* = 0.00), while the weakest correlation was registered in Group 3 (0.0969, *p* = 0.002).

There was no statistically significant correlation between the total number of suspect and concomitant drugs in non-serious ADR reports provided by HPs. By contrast, a statistically significant positive correlation was found in non-HP reports of non-serious ADRs, which may demonstrate lower engagement of professional healthcare representatives in non-serious ADR reporting. In serious ADR reports, a positive correlation was observed with a higher power for HPs than for non-HPs, which suggests a higher detail orientation and engagement of HPs in reporting serious ADRs. 

## 4. Discussion

The statistical analysis of our data demonstrated that the mean number of reactions in a single ICSR provided by non-HPs was higher than from HPs in the whole analysed group as well as in serious and non-serious ADR reports analysed separately. In the studied period and the reports on the analysed drugs, non-HPs reported more reactions but fewer medicinal products indicated as suspect or concomitant, whereas our analysis of correlations between HPs and non-HPs may suggest that severe adverse reactions are reported with more specificity by HPs (a higher number of reactions correlated with a higher number of reported drugs), regardless of whether the data was analysed for suspect count only or jointly as suspect and concomitant drug count. No statistically significant correlation between the total of suspect and concomitant drugs in non-serious ADR reports provided by HPs compared to the statistically significant positive correlation for reports of non-serious ADRs from non-HPs may indicate lower engagement of professional healthcare representatives in non-serious ADRs reporting. Weaker engagement of healthcare professionals in non-serious ADR reporting was observed based on correlation analysis for the reaction count and suspect count variables as well as for correlation analysis of reaction count and suspect and concomitant count variables, whereas in serious ADR reports, a higher level of detail orientation and engagement was observed.

The authors of this study indicate that pharmacovigilance cannot be based solely on patient reports since, in some cases, even life-threatening reactions were not connected with therapy and, consequently, were not reported by the patient. According to the authors, the coding system in consumer reports should take into account differences in language and terminology used by patients and healthcare professionals [21]. However, the results of another study indicated that patient reports enabled the identification of additional cases since some drug reactions that patients reported to the doctor were not confirmed and reported, and their occurrence was even negated despite scientific evidence for a connection between the symptoms and the medicinal product [22].

It is estimated that the rate of adverse reactions caused by interactions amounts to approximately 30% of all the reports filed [23]. The available scientific literature provides limited data on the use of databases for the collection and analysis of ADRs in order to identify and conduct research on drug interactions [24,25]. Spontaneous monitoring databases are mainly used to detect post-application ADRs, not drug combination-related reactions. There are, however, scientific studies that focus on the role of spontaneous monitoring and pharmacovigilance databases in establishing the incidence and characteristics of ADRs caused by drug interactions [26,27]. A study conducted by the Croatian HALMED agency, which used spontaneous monitoring data, confirmed that the database was a valuable source of data for detecting actual interactions between the drugs. For reported interactions, 38.7% of 1209 reports, including at least two drugs, concerned potential interactions, and 7.8% of all the analysed reports directly indicated drug interactions [27]. Our results consistently indicate that the number of reports with data on potential interactions may be higher than the reports that actually indicate their incidence. Some publications, however, indicate that potential interactions significantly exceed the number of actual interactions. According to the authors, statistical tools will not replace a correct assessment of risks and benefits made by a prescribing doctor while carefully monitoring patients [28]. 

Researchers also design statistical tools and models to detect drug interactions [23,29]. A decidedly lower number of available studies focused on the data from patients and their representatives in the detection of interactions and related risks. Our study concentrated on the differences and similarities between the data from non-HPs and HPs and thus on their importance in the detection of the safety of medicinal products, including potential NSAID-related interactions.

In the ‘primary drug’ group that was analysed, statistically significant differences in co-reporting were observed for naproxen (the differences reached 47 pp for the substances most frequently co-reported by both reporting groups), and the most substantial differences concerned proton pump inhibitors (K+/H+ ATPase), which are the top six substances on the list of 20 drugs most frequently co-reported with naproxen by non-HPs. The next most frequently co-reported substance from another group of drugs was acetylsalicylic acid, more frequently reported by the non-HPs group (15 percentage points). Further considerable differences were observed for ibuprofen and celecoxib. The former was more frequently co-reported by non-HPs with, i.a., chlorphenamine and pseudoephedrine (by 16.5 pp and 14.85 pp, respectively), whereas less frequently with rivaroxaban (by 14.72 pp). The slightest statistical differences in co-reporting concerned meloxicam and diclofenac. In the analysed ‘primary drug’ group, vitamins, including, i.a., vitamin D and ascorbic acid, were also more frequently co-reported by non-HPs. Non-HPs also more frequently reported co-use of other NSAID medications, including statins and levothyroxine.

The available literature and drug interaction databases describe, i.a., the following significant and most frequent NSAID interactions:increased risk of bleeding as a result of NSAIDs co-used with antithrombotic agents (such as warfarin),kidney failure as a result of NSAIDs combined with angiotensin-converting enzyme (ACE) inhibitors (used in cardiovascular diseases and hypertension treatment) and diuretics,decreased therapeutic effect of medications for heart failure and hypertension, including ACE inhibitors, beta-blockers, and diuretics, when co-used with NSAIDs,increased risk of gastrointestinal ulceration or bleeding as a result of different types of NSAID combinations (including low-dose aspirin) or NSAIDs co-used with corticosteroids [30].

According to the available studies, gastrointestinal tract-related ADRs were the most common NSAID-induced adverse reactions; such symptoms were reported by as many as 60–70% of patients [31]. NSAID use also poses a risk of NSAID-associated AKI and interactions primarily with diuretics and renin–angiotensin–aldosterone system inhibitors [32,33]. 

Moreover, the results of our analysis indicated that kidney failure and increased risk of bleeding, including gastrointestinal bleeding, were considerably more frequently reported by HPs than non-HPs, and they are common NSAID-related adverse reactions. These reactions may also accompany rare interactions with NSAIDs. Non-HPs, in turn, more frequently reported hypersensitive responses and allergies to NSAIDs, as well as all other reactions that impact the quality of life, including the ineffectiveness of therapy, which may commonly result in reported pains. O ur analyses indicate that non-HPs most frequently report the reactions that may be considered less significant by HPs, do not meet the criteria of reactions to medication (product use in an unapproved indication, off-label use, drug ineffectiveness), or have a more significant impact on the patient’s quality of life (pain, headache, arthralgia, peripheral swelling, fatigue).

Contrastingly, the reactions most frequently reported by HPs are not on the list of 20 drugs most frequently reported by non-HPs, such as gastrointestinal haemorrhage, AKI, melaena, upper gastrointestinal haemorrhage, anaemia, or indications of a drug interaction. Such reactions require a professional medical diagnosis on the basis of other symptoms described by the patients or additional diagnostic procedures.

Another challenge is how to minimise newly detected as well as already increasing risks caused by drug–drug interactions. For OTC drugs, the problem is particularly important due to an increasingly common self-treatment problem. Moreover, the information included in patient information leaflets, especially about interactions, contraindications, dosage instructions, and side effects, is often too complex for elderly people and those with low literacy skills. Despite the action taken to enhance the readability of patient information leaflets by means of templates and user testing, sometimes additional risk minimization still appears necessary [34]. Updated therapeutic leaflets, short booklets on drug interactions attached to the leaflet, and promotion of patient access to websites discussing drug interactions can help resolve the identified problem and reduce the issue of drug–drug interactions [35].

## 5. Conclusions

In the studied period and the reports provided by the analysed groups, non-HPs reported a higher number of reactions but a lower number of medications indicated as suspect or concomitant. Although HPs reported fewer reactions on average, their diversity was greater compared to non-HP reports. An analysis of the correlation of the number of reactions with the number of drugs reported in a single case may demonstrate the detail orientation and higher engagement of HPs in reporting serious adverse reactions. On the other hand, our correlation analysis showed that non-serious adverse reaction reports submitted by non-HPs were more detailed when compared to HPs.

The results of our analysis indicate that kidney failure and increased risk of bleeding, including gastrointestinal bleeding, were much more frequently reported by HPs compared to non-HPs, and these reactions are known and described in the literature as NSAID-related ADRs as well as common symptoms of NSAID interaction. Non-HPs more frequently reported reactions that may be considered less significant by HPs, do not meet the criteria of reactions to medication (product use in an unapproved indication, off-label use, and drug ineffectiveness), or describe a significant impact on the patient’s quality of life (pain, headache, arthralgia, peripheral swelling, and fatigue). The differences in reports may result, i.a., from the fact that the reports from the patients and their caregivers require a professional medical diagnosis established on the basis of the symptoms described by the patient or additional diagnostic tests. This implies that, when appropriately classified, medically verified, and statistically analysed, the collected evidence may even provide new data on drug interactions or symptoms, specifically once they have been confirmed by healthcare professionals.

## Figures and Tables

**Figure 1 medicina-59-01585-f001:**
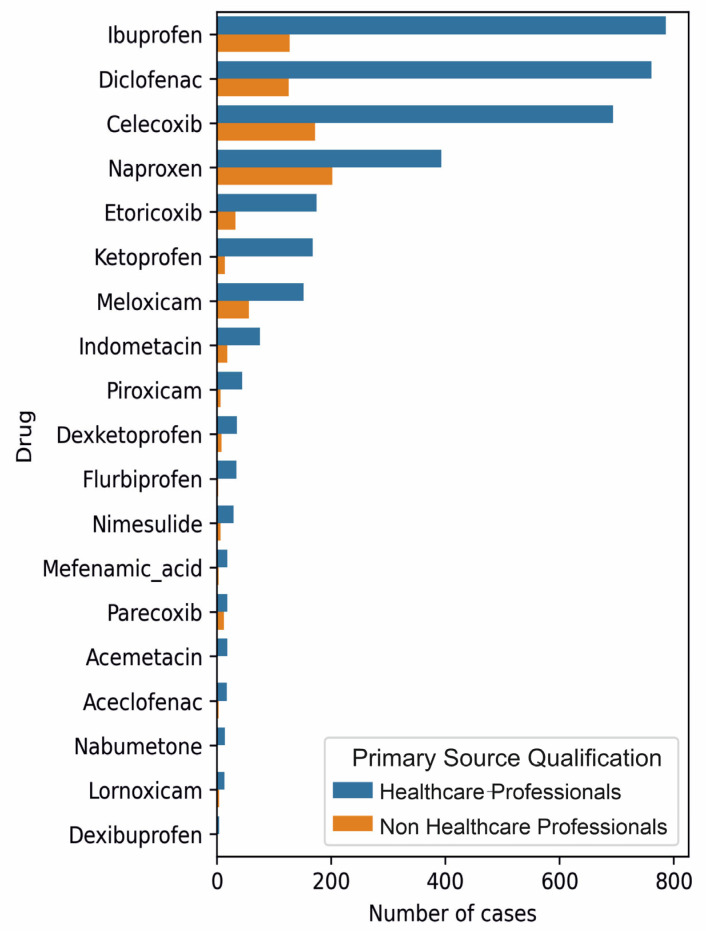
The number of cases of specific primary drugs with the total of suspect and concomitant drugs equal to or higher than 5.

**Figure 2 medicina-59-01585-f002:**
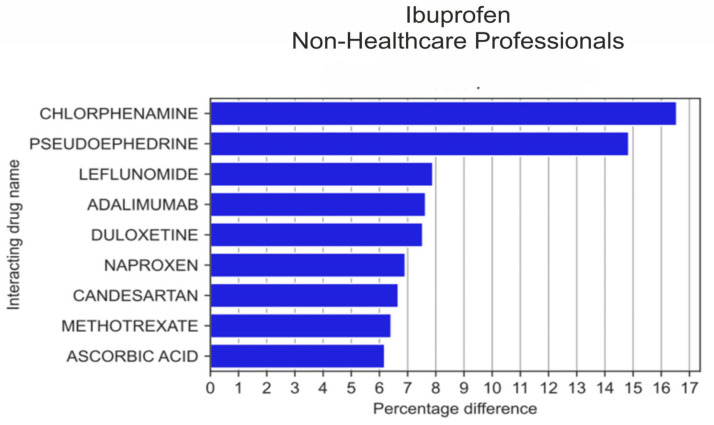
Statistically significant percentage difference in drug reporting ratio from the group of 20 drugs most frequently co-reported with ibuprofen by non-HPs.

**Figure 3 medicina-59-01585-f003:**
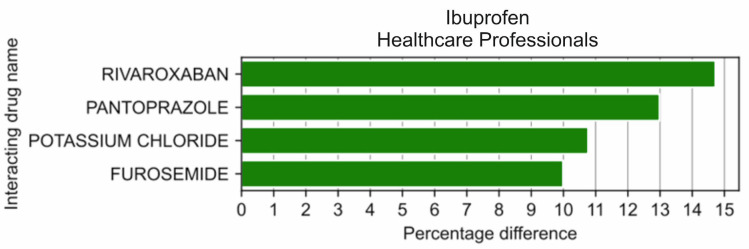
Statistically significant percentage difference in drug reporting ratio from the group of 20 drugs most frequently co-reported with ibuprofen by HPs.

**Figure 4 medicina-59-01585-f004:**
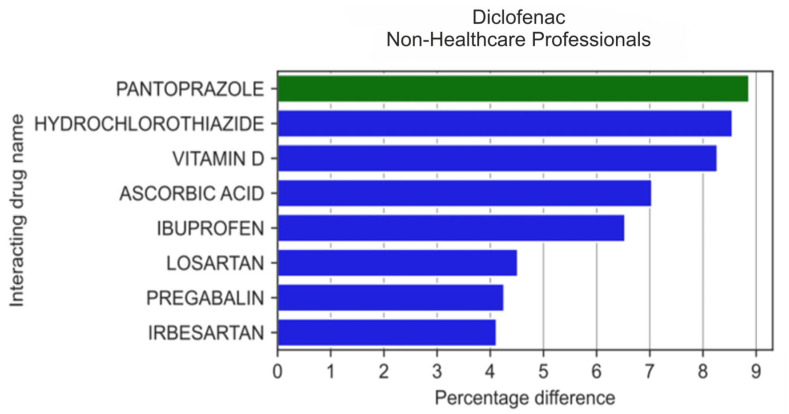
Statistically significant percentage difference in drug reporting ratio from the group of 20 drugs most frequently co-reported with diclofenac by non-HPs.

**Figure 5 medicina-59-01585-f005:**
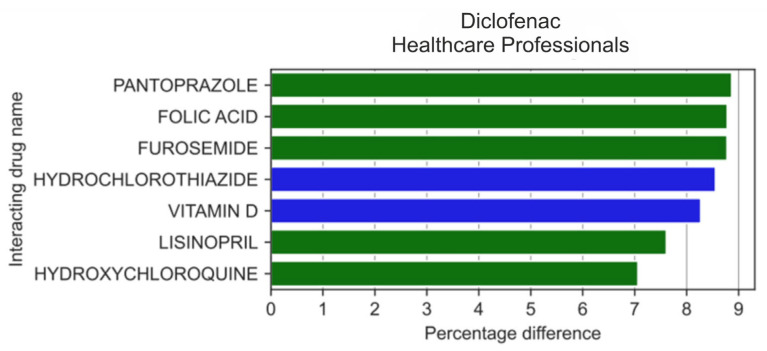
Statistically significant percentage difference in drug reporting ratio from the group of 20 drugs most frequently co-reported with diclofenac by HPs.

**Figure 6 medicina-59-01585-f006:**
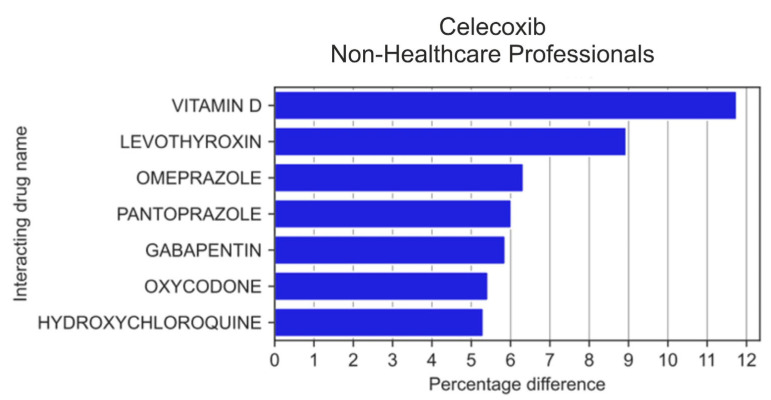
Statistically significant percentage difference in drug reporting ratio from the group of 20 drugs most frequently co-reported with celecoxib by non-HPs.

**Figure 7 medicina-59-01585-f007:**
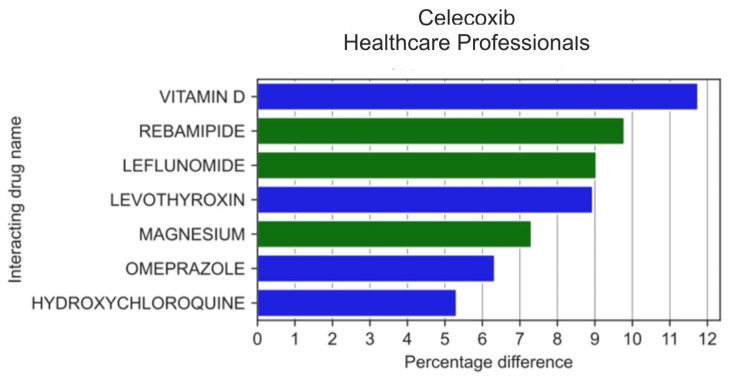
Statistically significant percentage difference in drug reporting ratio from the group of 20 drugs most frequently co-reported with celecoxib by HPs.

**Figure 8 medicina-59-01585-f008:**
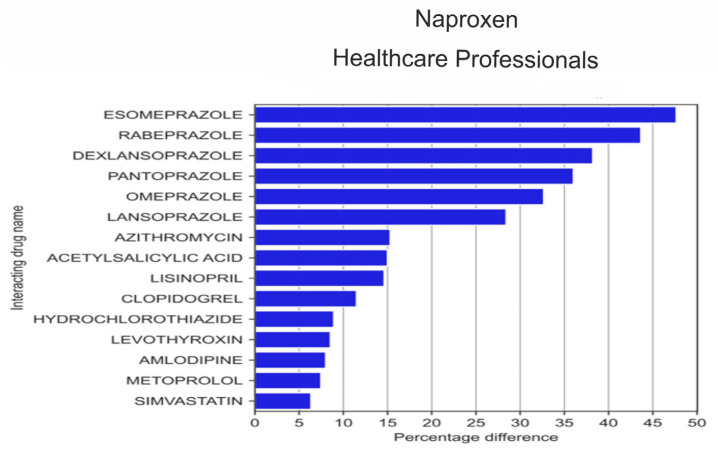
Statistically significant percentage difference in drug reporting ratio from the group of 20 drugs most frequently co-reported with naproxen by HPs.

**Figure 9 medicina-59-01585-f009:**
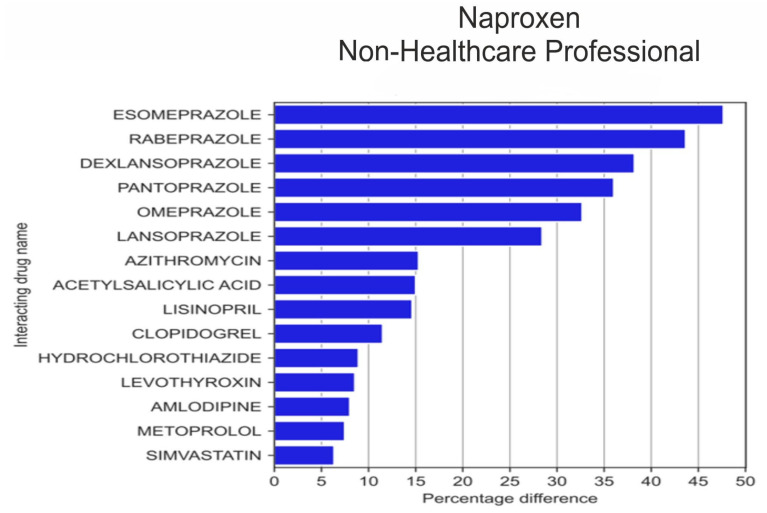
Statistically significant percentage difference in drug reporting ratio from the group of 20 drugs most frequently co-reported with naproxen by non-HPs.

**Figure 10 medicina-59-01585-f010:**
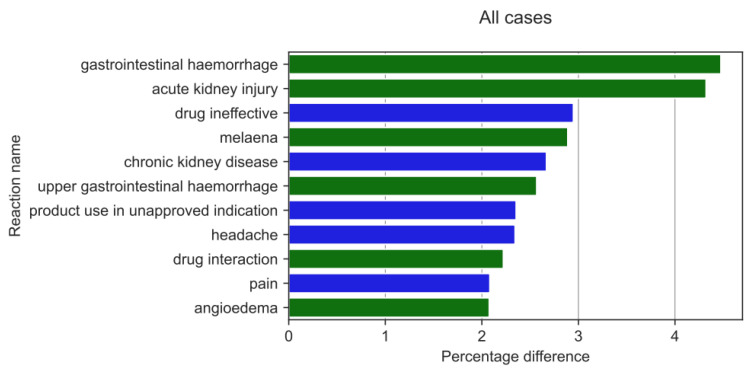
The highest statistically significant percentage difference in reaction reporting ratio (reaction reported more frequently by HPs in green colour and by non-HPs in blue colour).

**Figure 11 medicina-59-01585-f011:**
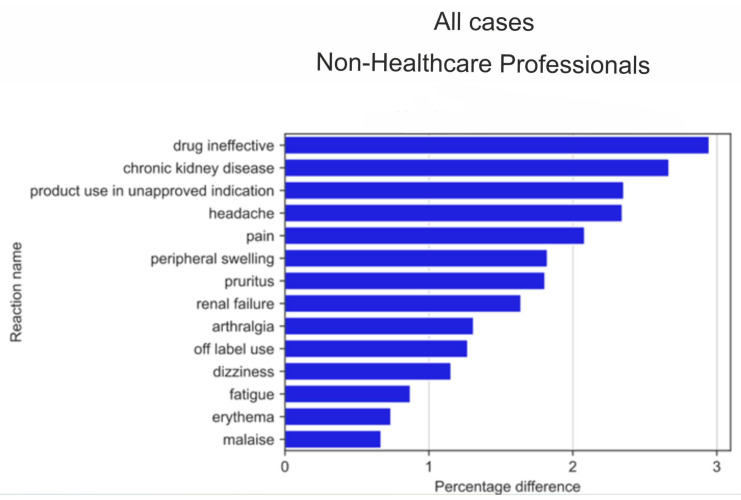
Statistically significant percentage difference in reaction reporting ratio from the group of 20 most frequent drug reactions reported for the analysed group of substances by non-HPs (reaction reported more frequently by non-HPs in blue colour).

**Figure 12 medicina-59-01585-f012:**
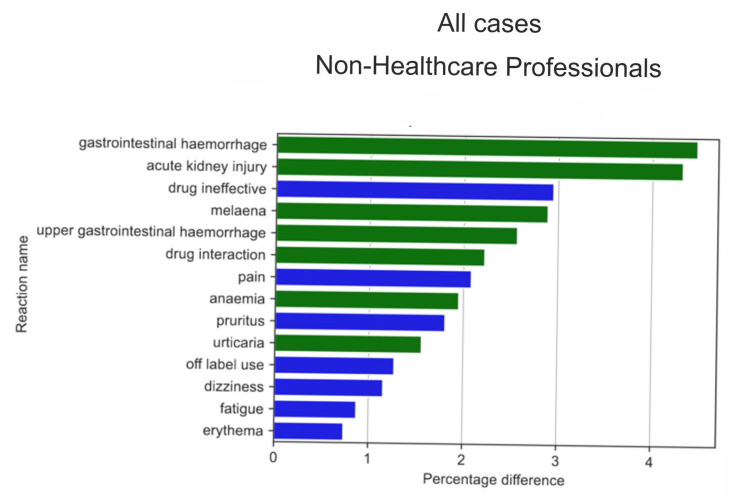
Statistically significant percentage difference in reaction reporting ratio from the group of 20 most frequent drug reactions reported for the analysed group of substances by HPs (reaction reported more frequently by HPs in green colour and by non-HPs in blue colour).

**Figure 13 medicina-59-01585-f013:**
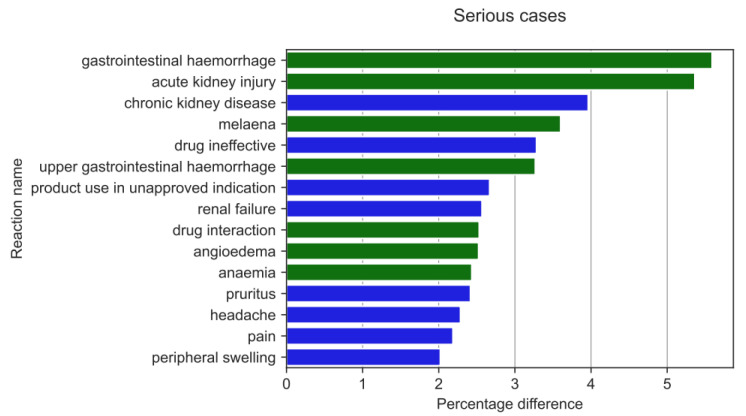
The highest statistically significant percentage difference in serious reaction reporting ratio (reaction reported more frequently by HPs in green colour and by non-HPs in blue colour).

**Figure 14 medicina-59-01585-f014:**
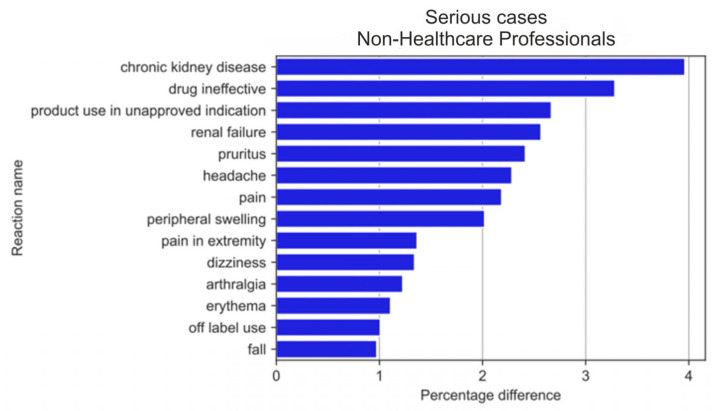
Statistically significant percentage difference in reaction reporting ratio from the group of 20 most frequent drug reactions reported in serious cases for the analysed group of substances (reaction reported more frequently by non-HPs in blue colour).

**Figure 15 medicina-59-01585-f015:**
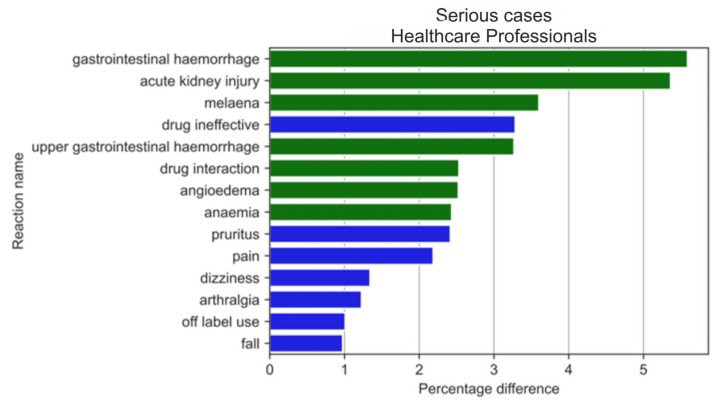
Statistically significant percentage difference in reaction reporting ratio from the group of 20 most frequent drug reactions reported in serious cases for the analysed group of substances by HPs (reaction reported more frequently by HPs in green colour and by non-HPs in blue colour).

**Figure 16 medicina-59-01585-f016:**
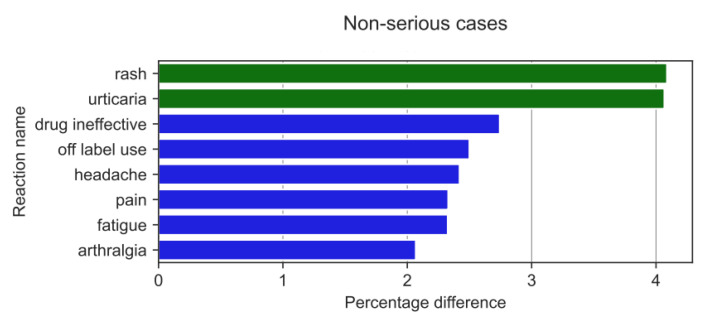
Statistically significant percentage difference in reaction reporting ratio from the group of 20 most frequent drug reactions reported in non-serious cases for the analysed group of substances by HPs (reaction reported more frequently by HPs in green colour and by non-HPs in blue colour).

**Figure 17 medicina-59-01585-f017:**
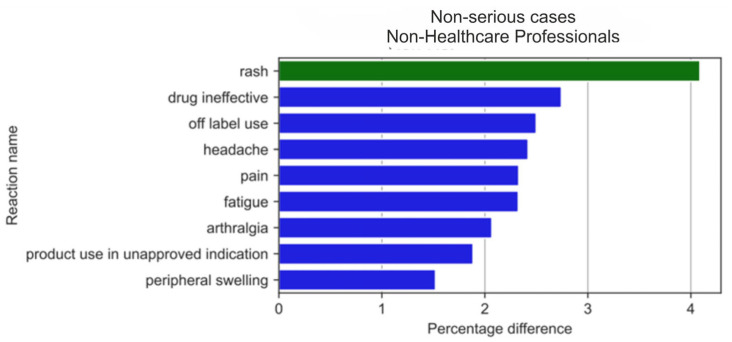
Statistically significant percentage difference in reaction reporting ratio for the group of 20 most frequent drug reactions reported in non-serious cases for the analysed group of substances by non-HPs (reaction reported more frequently by HPs in green colour and by non-HPs in blue colour).

**Figure 18 medicina-59-01585-f018:**
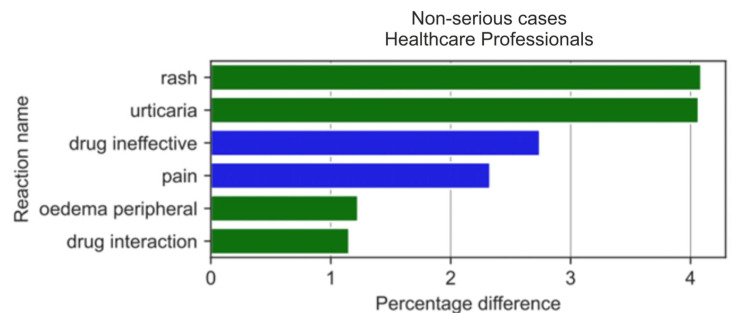
Statistically significant percentage difference in reaction reporting ratio for the group of 20 most frequent drug reactions reported in non-serious cases for the analysed group of substances by HPs (reaction reported more frequently by HPs in green colour and by non-HPs in blue colour).

**Figure 19 medicina-59-01585-f019:**
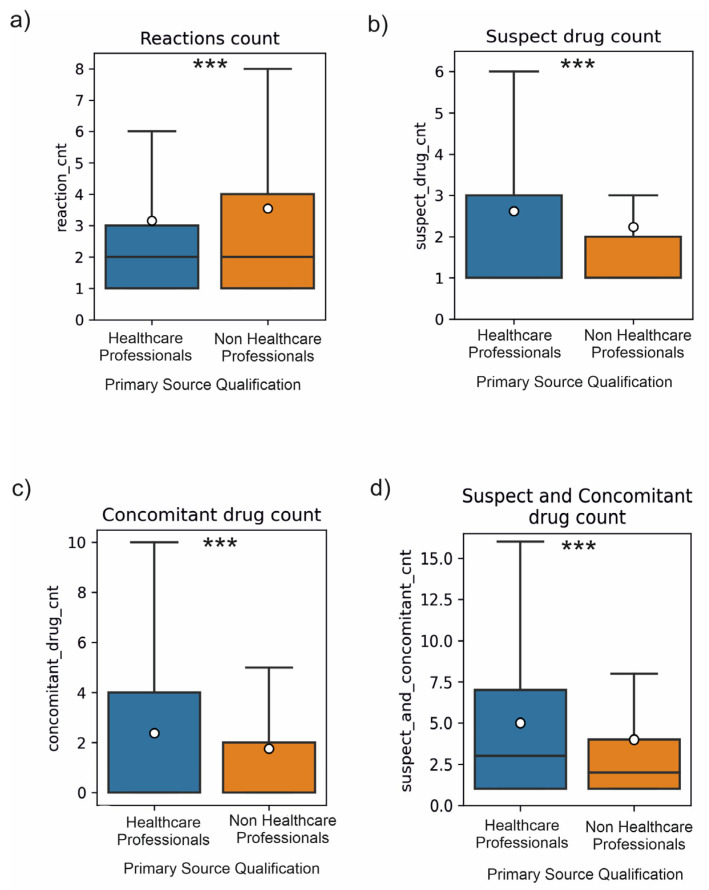
Comparison of the observed distribution of data on the number of ‘reactions’, ‘suspect’, ‘concomitant’, and ‘suspect and concomitant’ drugs. Box-and-whisker plots showing the number of reported (**a**) reactions (**b**) suspect drugs (**c**) concomitant drugs (**d**) suspect and concomitant drugs by Healthcare professionals or non-Healthcare professionals. The white circle represents the mean. The statistical analysis was performed using the Mann–Whitney test. The *p*-values were corrected for multiple comparisons using the Bonferroni method. *** *p*-value < 0.001.

**Table 1 medicina-59-01585-t001:** Data on the mean number of reactions, mean number of suspected drugs, and concomitant drugs reported between 2018 and 2021 by Healthcare Professionals (HPs) and non-Healthcare Professionals (non-HPs).

Non-Serious Adverse Drug Reaction Reports																																
	**reaction_cnt**								**suspect_drug_cnt**								**concomitant_drug_cnt**								**suspect_and_concomitant_cnt**							
	count	Mean	std	Min	25%	50%	75%	Max	Count	Mean	std	min	25%	50%	75%	Max	Count	Mean	Std	Min	25%	50%	75%	max	Count	mean	Std	min	25%	50%	75%	Max
HPs	2193	2.03	1.40	1	1	2	3	17	2193	1.39	1.30	1	1	1	1	25	2193	1.50	2.87	0	0	0	2	29	2193	2.89	3.16	1	1	1	4	30
Non-HPs	1049	2.56	1.94	1	1	2	3	18	1049	1.31	0.86	1	1	1	1	13	1049	0.98	1.80	0	0	0	1	20	1049	2.29	2.04	1	1	1	3	21
Serious adverse drug reaction reports																																
	**reaction_cnt**								**suspect_drug_cnt**								**concomitant_drug_cnt**								**suspect_and_concomitant_cnt**							
	**Count**	**Mean**	**std**	**Min**	**25%**	**50%**	**75%**	**Max**	**count**	**Mean**	**std**	**Min**	**25%**	**50%**	**75%**	**max**	**Count**	**Mean**	**Std**	**Min**	**25%**	**50%**	**75%**	**Max**	**Count**	**mean**	**Std**	**min**	**25%**	**50%**	**75%**	**Max**
HPs	7183	3.50	4.90	1	1	2	4	61	7183	3.00	3.84	1	1	2	3	42	7183	2.64	4.26	0	0	0	4	47	7183	5.64	6.12	1	2	4	8	63
Non-HPs	2317	4.00	4.59	1	1	3	5	55	2317	2.65	3.20	1	1	1	3	46	2317	2.10	4.59	0	0	0	2	51	2317	4.75	6.52	1	1	2	5	55
All adverse drug reaction reports																																
	**reaction_cnt**								**suspect_drug_cnt**								**concomitant_drug_cnt**								**suspect_and_concomitant_cnt**							
	**Count**	**Mean**	**std**	**min**	**25%**	**50%**	**75%**	**Max**	**count**	**Mean**	**std**	**min**	**25%**	**50%**	**75%**	**Max**	**Count**	**Mean**	**Std**	**Min**	**25%**	**50%**	**75%**	**Max**	**Count**	**mean**	**Std**	**min**	**25%**	**50%**	**75%**	**Max**
HPs	9376	3.16	4.39	1	1	2	3	61	9376	2.62	3.48	1	1	1	3	42	9376	2.38	4.01	0	0	0	4	47	9376	5.00	5.69	1	1	3	7	63
Non-HPs	3366	3.55	4.01	1	1	2	4	55	3366	2.23	2.77	1	1	1	2	46	3366	1.75	3.97	0	0	0	2	51	3366	3.98	5.64	1	1	2	4	55

**Table 2 medicina-59-01585-t002:** The results of a post-hoc analysis conducted for the reaction_cnt variable.

Group		RIR Tukey’s Test; Variable: reaction_cnt Marked Differences are Significant with *p* < 0.05000			
		1	2	3	4
		M = 2.0333	M = 3.4981	M = 2.5624	M = 3.9974
1	1		0.000008	0.005032	0.000008
2	2	0.000008		0.000008	0.000012
3	3	0.005032	0.000008		0.000008
4	4	0.000008	0.000012	0.000008	

**Table 3 medicina-59-01585-t003:** The results of a post-hoc analysis conducted for the suspect_drug_cnt variable.

Group		RIR Tukey’s Test; Variable: suspect_drug_cnt Marked Differences are Significant with *p* < 0.05000			
		1	2	3	4
		M = 1.3881	M =2.9993	M = 1.3108	M = 2.6526
1	1		0.000008	0.920817	0.000008
2	2	0.000008		0.000008	0.000051
3	3	0.920817	0.000008		0.000008
4	4	0.000008	0.000051	0.000008	

## Data Availability

Data are available in a publicly accessible repository. The data presented in this study are openly available on EudraVigilance—the European database of suspected adverse drug reaction reports, at www.adrreports.eu (accessed on 13 April 2022).

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
