# Peer review of "Indirect and Direct 65+ Patient Reporting of Non-Steroidal Anti-Inflammatory Drug-Induced Adverse Drug Reactions as a Source of Information on Polypharmacy and Polypharmacy-Related Risk"

_medicina, 2023, doi:10.3390/medicina59091585_

Round 1
Reviewer 1 Report
In light of how frequently NSAIDS therapy is used by elderly patients, this research addresses an important topic.
For Eudravigilance only the doctors enrroled the adverse events, or even the patients? ( Non HP)
It is not very clear for me how was defined by the authors the terms: , ‘suspect’, ‘concomitant’ and ‘suspect and concomitant’ drugs.
For the Discussion session :Updated therapeutic leaflets, short booklets with drug interactions attached to the leaflet, and possibly patient access to websites with drug interactions are all necessary for solving the alarm raised by the article.
-
Author Response
Dear Sir/Madam,
We wish to express our appreciation for the comments and suggestions for our manuscript entitled “Indirect and Direct 65+ Patient Reporting of Non-steroidal Anti-inflammatory Drugs-induced Adverse Drug Reactions as Source of Information on Polypharmacy and Polypharmacy-related risk”. We have carefully revised the manuscript taking into consideration all the comments.
Kind regards,
Kamila Sienkiewicz

Reviewer 2 Report
The article “Indirect and Direct 65+ Patient Reporting of Non-steroidal 2 Anti-inflammatory Drugs-induced Adverse Drug Reactions as 3 Source of Information on Polypharmacy and Polypharmacy-related risk” provides an information of statistically analyzed method to de-risks non-steroidal anti-inflammatory drugs-induced adverse drug reactions.
Comments to authors:
1) The author may add information related to the population ethnicity or gender in the manuscript. Also, discuss whether this method will be applicable to the wide population or not.
2) I found the result section very monotonous and same language in most of result section. The author may change the language bit and make it interesting for the reader.
3) There are a few grammatical errors in the manuscript. Please, proofread the revised version.
Minor revision required
Author Response

(The authors gave the same response as above.)
